# Conditional Expectation based Value Decomposition for Scalable On-Demand Ride Pooling

## Abstract

Owing to the benefits for customers (lower prices), drivers (higher revenues), aggregation companies (higher revenues) and the environment (fewer vehicles), on-demand ride pooling (e.g., Uber pool, Grab Share) has become quite popular. The significant computational complexity of matching vehicles to combinations of requests has meant that traditional ride pooling approaches are myopic in that they do not consider the impact of current matches on future value for vehicles/drivers. Recently, Neural Approximate Dynamic Programming (NeurADP) has employed value decomposition with Approximate Dynamic Programming (ADP) to outperform leading approaches by considering the impact of an individual agent's (vehicle) chosen actions on the future value of that agent. However, in order to ensure scalability and facilitate city-scale ride pooling, NeurADP completely ignores the impact of other agents actions on individual agent/vehicle value. As demonstrated in our experimental results, ignoring the impact of other agents actions on individual value can have a significant impact on the overall performance when there is increased competition among vehicles for demand. Our key contribution is a novel mechanism based on computing conditional expectations through joint conditional probabilities for capturing dependencies on other agents actions without increasing the complexity of training or decision making. We show that our new approach, Conditional Expectation based Value Decomposition (CEVD) outperforms NeurADP by up to 9.76% in terms of overall requests served, which is a significant improvement on a city wide benchmark taxi dataset.

## 1 Introduction

Taxi/car on Demand (ToD) services (e.g., UberX, Lyft, Grab) not only provide a comfortable means of transport for customers, but also are good for the environment by enabling sharing of vehicles over time (while being used to serve one request at any one point in time). A further improvement of ToD is on-demand ride pooling (e.g., UberPool, LyftLine, GrabShare etc.), where vehicles are shared not only over time but also in space (on the taxi/car). On-demand ride pooling reduces the number of vehicles required, thereby reducing emissions and traffic congestion compared to Taxi/car on-Demand (ToD) services. This is achieved while providing benefits to all the stakeholders involved: (a) Individual passengers have reduced costs due to sharing of space; (b) Drivers make more money per trip as multiple passengers (or passenger groups) are present; (c) For the aggregation company more customer requests can be satisfied with the same number of vehicles.

In this paper, we focus on this on-demand ride pooling problem at city scale, referred to as Ride-Pool Matching Problem (RMP) [Alonso-Mora et al. (2017); Bei & Zhang (2018); Lowalekar et al. (2019)]. The goal in an RMP is to assign combinations of user requests to vehicles (of arbitrary capacity) online such that quality constraints (e.g., delay in reaching destination due to sharing is not more than 10 minutes) and matching constraints (one request can be assigned at most one vehicle, one vehicle must be assigned at most one request combination) are satisfied while maximizing an overall objective (e.g., number of requests, revenue). Unlike the ToD problem that requires solving a bipartite matching problem between vehicles and customers, RMP requires effective matching on a tripartite graph of requests, trips (combinations of requests) and vehicles. This matching on tripartite graph significantly increases the complexity of solving RMP online, especially at city scale where

there are hundreds or thousands of vehicles, hundreds of requests arriving every minute and request combinations have to be computed for each vehicle.

Due to this complexity and the need to make decisions online, most existing work related to solving RMP has focused on computing best greedy assignments [Ma et al. (2013); Tong et al. (2018); Huang et al. (2014); Lowalekar et al. (2019); Alonso-Mora et al. (2017)]. While these scale well, they are myopic and, as a result, do not consider the impact of a given assignment on future assignments. The closest works of relevance to this paper are by Shah *et al.* [Shah et al. (2020)] and Lowalekar *et al.* [Lowalekar et al. (2021)]. We specifically focus on the work by Shah *et al.*, as it has the best performance, while being scalable. That work considers future impact of current assignment from an individual agents' perspective without sacrificing on scalability (to city scale). However, a key limitation of that work is that they do not consider the impact of other agents (vehicles) actions on an agents'(vehicle) future impact, which as we demonstrate in our experiments can have a major effect (primarily because vehicles are competing for the common demand).

To that end, we develop a conditional expectation based value decomposition approach that not only considers future impact of current assignments but also of other agents state and actions through the use of conditional probabilities and tighter estimates of individual impact. Due to these conditional probability based tighter estimates of individual value functions, we can scale the work by Guestrin *et al.* [Guestrin & Parr (2002)] and Li *et al.* [Li & Kochenderfer (2021)] to solve problems with no explicit coordination graphs and hundreds/thousands of homogeneous agents. Unlike value decomposition approaches [Rashid & Whiteson (2018); Sunehag & Graepel (2018)] developed for solving cooperative Multi-Agent Reinforcement Learning (MARL) with tens of agents and under centralized training and decentralized execution set up, we focus on problems with hundreds or thousands of agents with centralized training and centralized execution (e.g., Uber, Lyft, Grab).

In this application domain of taxi on demand services, where improving 0.5%-1% is a major achievement [Lin et al. (2018)], we demonstrate that our approach easily outperforms the existing best approach, NeurADP [Shah et al. (2020)] by at least 3.8% and up to 9.76% on a wide variety of settings for the benchmark real world taxi dataset [NYYellowTaxi (2016)].

## 2 BACKGROUND

In this section, we formally describe the RMP problem and also provide details of an existing approach for on-demand ride pooling called NeurADP, which we improve over.

**Ride-pool Matching Problem (RMP)** : We consider a fleet of vehicles/resources $\mathcal{R}$ with random initial locations, travelling on a predefined road network $\mathcal{G}$ with intersections : $\mathcal{L}$ as nodes, road segments : $\mathcal{E}$ as edges and weights on edges indicate the travel time on the road segment. Passengers that want to travel from one location to another send requests to a central entity that collects these requests over a time-window called the decision epoch $\Delta$. The goal of the RMP is to match these collected requests $\mathcal{U}^t$ to empty or partially filled vehicles that can serve them such that an objective $\mathcal{J}$ is maximised subject to constraints on the delay $\mathcal{D}$.

We upperbound $\mathcal{D}$ and consider the objective $\mathcal{J}$ to be the number of requests served. Thus, RMP is defined using the tuple $[\mathcal{G}, \mathcal{U}, \mathcal{R}, \mathcal{D}, \Delta, \mathcal{J}]$[1]. Please refer Appendix A.1 for a detailed description.

Delay constraints : $\mathcal{D}$ consider two delays, $\{\tau, 2\tau\}$. $\tau$ denotes the maximum allowed pick-up delay which is the difference between the arrival time of a request and the time at which a vehicle picks the user up. $2\tau$ denotes the maximum allowed detour delay which is the difference between the time at which the user arrived at their destination in a shared cab and the time at which they would have arrived if they had taken a single-passenger cab.

**Neural Approximate Dynamic Programming (NeurADP) for Solving RMP:** Figure 1 provides the overall approach. In this paper, there are two NeurADP [Shah et al. (2020)] contributions of relevance:

> **FV:** *To estimate **F**uture **V**alue of current actions, a method for solving the underlying Approximate Dynamic Program (ADP) [Powell (2007)] by considering neural network representations of value functions.*

---

[1]Everywhere in the paper $[,]$ is used as the concatenation operator

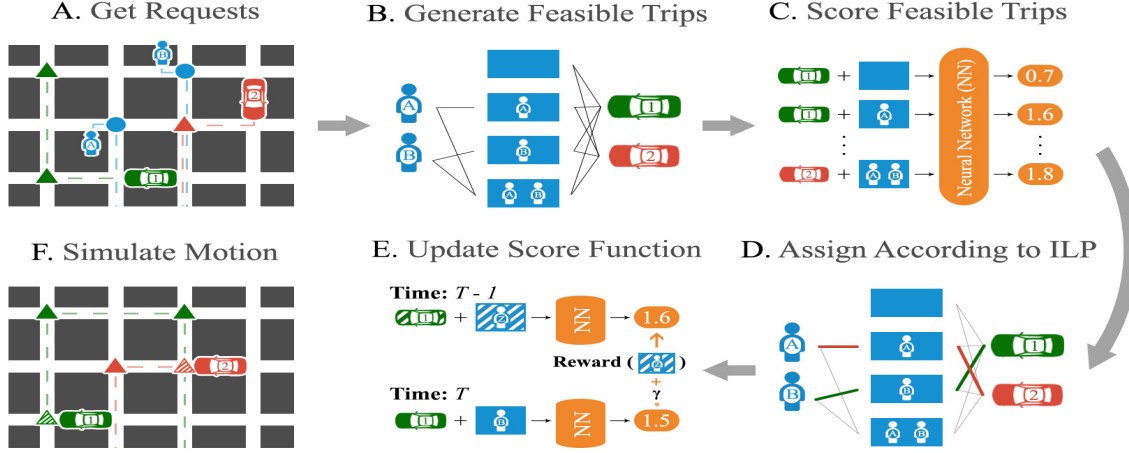

Figure 1: NeurADP approach [Shah et al. (2020)]

***DJV:*** *To ensure scalability, **D**ecomposing the **J**oint **V**alue function into individual vehicle value functions by extending on the work of Russell et al. [Russell & Zimdars (2003)].*

**F**uture **V**alue (**FV**): ADP is similar to a Markov Decision Problem (MDP) with the key difference that the transition uncertainty is extrinsic to the system and not dependent on the action. The ADP problem for RMP is formulated using the tuple $\langle S, A, \xi, T, \mathcal{J} \rangle$, where :

$S$ : The state of the system is represented as $s_t = (r_t, u_t)$ where $r_t$ is the state of all vehicles and $u_t$ contains all the requests waiting to be served. The state is obtained in Step A of Figure 1.

$A$ : At each time step there are a large number of requests arriving to the taxi service provider, however for an individual vehicle only a small number of such requests are reachable. The feasible set of request combinations for each vehicle $i$ at time $t$, $\mathcal{F}_t^i$ is computed in Step B of Figure 1:

$$\mathcal{F}_t^i = \{f^i | f^i \in \cup_{c'=1}^{c^i}[\mathcal{U}]^{c'}, \text{PickUpDelay}(f^i, i) \leq \tau, \text{DetourDelay}(f^i, i) \leq 2\tau\} \quad (1)$$

$a_t^{i,f}$ is the decision variable that indicates whether vehicle $i$ takes action $f$ (a combination of requests) at a decision epoch $t$. Joint actions across vehicles have to satisfy matching constraints: (i) each vehicle, $i$ can only be assigned at most one request combination, $f$; (ii) at most one vehicle, $i$ can be assigned to a request $j$; and (iii) a vehicle, $i$ can be either assigned or not assigned to a request combination.

$$\sum_{f \in \mathcal{F}_t^i} a_t^{i,f} = 1 ::: \forall i \in \mathcal{R} \quad \sum_{i \in \mathcal{R}} \sum_{f \in \mathcal{F}_t^i; j \in f} a_t^{i,f} \leq 1 ::: \forall j \in \mathcal{U}_t \quad a_t^{i,f} \in \{0, 1\} ::: \forall i, f \quad (2)$$

$\xi$ : denotes the exogenous information – the source of randomness in the system. This would correspond to the user requests or demand. $\xi_t$ denotes the exogenous information at time $t$.

$T$ : denotes the transitions of system state. In an ADP, the system evolution happens as $(s_0, a_0, s_0^a, \xi_1, s_1, a_1, s_1^a, \cdots, s_t, a_t, s_t^a, \cdots)$, where $s_t$ denotes the pre-decision state at decision epoch $t$ and $s_t^a$ denotes the post-decision state [Powell (2007)]. The transition from state $s_t$ to $s_{t+1}$ depends on the action vector $a_t$ and the exogenous information $\xi_{t+1}$. Therefore,

$$s_{t+1} = T(s_t, a_t, \xi_{t+1}); s_t^a = T^a(s_t, a_t); s_{t+1} = T^\xi(s_t^a, \xi_{t+1})$$

It should be noted that $T^a(., .)$ is deterministic as uncertainty is extrinsic to the system.

$\mathcal{J}$ : denotes the reward function and in RMP, this will be the revenue from a trip.

Let $V(s_t)$ denotes the value of being in state $s_t$ at decision epoch $t$, then using Bellman equation:

$$V(s_t) = \max_{a_t \in A_t} (\mathcal{J}(s_t, a_t) + \gamma \mathbb{E}[V(s_{t+1})|s_t, a_t, \xi_{t+1}]) \quad (3)$$

**(I) Get Requests, Compute Feasible Actions, Find Individual Values**

**(II) Score Values considering consequences on other vehicles**

$$\hat{V}_{2|f_1} = pV_2(g_1) + (1-p)V_2(g_2)$$
[Expected Value of $R_2$ conditioned on $f_1$]

$$\hat{V}_1(f_1) = wV_1(f_1) + (1-w)\hat{V}_{2|f_1}$$
[Neighbour aware value for action $f_1$ for $R_1$]

$$\hat{V}_{2|f_2} = qV_2(g_1) + (1-q)V_2(g_2)$$
[Expected Value of $R_2$ conditioned on $f_2$]

$$\hat{V}_1(f_2) = wV_1(f_2) + (1-w)\hat{V}_{2|f_2}$$
[Neighbour aware value for action $f_2$ for $R_1$]

**(III)Solve ILP using Neighbour aware scores, assign optimal actions, simulate motion**

Figure 2: Schematic outlining CEVD's neighbour aware scoring mechanism

where $\gamma$ is the discount factor. Using post-decision state, this expression breaks down nicely:

$$V(s_t) = \max_{a_t \in A_t} (\mathcal{J}(s_t, a_t) + \gamma V^a(s_t^a)); \qquad V^a(s_t^a) = \mathbb{E}[V(s_{t+1})|s_t^a, \xi_{t+1}] \qquad (4)$$

The advantage of this two step value estimation is that the maximization problem in Equation 4 can be solved using a Integer Linear Program (ILP) with matching constraints indicated in expression 2. Step D of Figure 1) provides this aspect of the overall algorithm. The value function approximation around post-decision state, $V^a(s_t^a)$ is a neural network and is updated (Step E of Figure 1) by stepping forward through time using sample realizations of exogenous information (i.e. demand observed in data). However, as we describe next, maintaining a joint value function is not scalable and hence we decompose and maintain individual value functions.

**Decomposing Joint Value (DJV)**: Non-linear value functions, unlike their linear counterparts, cannot be directly integrated into the ILP mentioned above. One way to incorporate them is to evaluate the value function for all possible post-decision states and then add these values as constants. However, the number of post-decision states is exponential in the number of resources/vehicles.

[Shah et al. (2020)] introduced a two-step decomposition of the joint value function that converts it into a linear combination over individual value functions associated with each vehicle. In the first step, following [Russell & Zimdars (2003)], the joint value function is written as the sum over individual value functions : $V(s_t^a) = \sum_i V^i(s_t^a)$.

In the second step, the individual vehicles' value functions are approximated. They assumed that the long-term expected reward of a given vehicle is not significantly affected by the specific actions another vehicle makes in the current decision epoch and thereby completely neglect the impact of the actions taken by other vehicles at the current time step. Thus they model the value function using the pre-decision, rather than post-decision, state of other vehicles which gives :

$$V^i(s_t^a) = V^i([s_t^{i,a}, s_t^{-i,a}]) \approx V^i([s_t^{i,a}, s_t^{-i}])$$

where $-i$ refers to all vehicles except vehicle $i$. This allows NeurADP to get around the *combinatorial explosion of the post-decision state of all vehicles*. NeurADP thus has the joint value function : $V(s_t^a) = \sum_i V^i([s_t^{i,a}, s_t^{-i}])$.

They then evaluate these individual $V^i$ values (Step C of Figure 1) for all possible $s_t^{i,a}$ (from the individual value neural network) and then integrate the overall value function into the ILP as a linear function over these individual values. This reduces the number of evaluations of the non-linear value function from exponential to linear in the number of vehicles.

## 3 CONDITIONAL EXPECTATION BASED VALUE DECOMPOSITION, CEVD

One of the fundamental drawbacks in NeurADP is that each agent/vehicle[2] to a large extent is kept in oblivion about the values of the feasible actions for other agents/vehicles. Since our problem execution (assignment of requests to agents) is centralized, this independence of individual agents (as shown in experimental results) leads to sub-optimal actions for the entire system.

While there are dependencies between agents, not all agents are dependent on each other and one mechanism typically employed to represent sparsely connected multi-agent systems is through the use of a coordination graph [ Guestrin & Parr (2002); Li & Kochenderfer (2021)], $CG = (X, E)$. The joint value of the system with joint state $s$ and joint action $a$ in the context of a coordination graph is given by:

$$Q^{CG}(s,a) = \sum_{i \in X} Q_i^{CG}(s,a); \qquad Q_i^{CG}(s,a) = f^i(a^i|s) + \sum_{j|(i,j) \in E} f^{ij}(a^i, a^j|s) \qquad (5)$$

where $f^i(.|.)$ represents the value of agent $i$ and $f^{ij}(.,.|.)$ represents the impact of agent $j$'s actions on agent $i$'s value. Such an approach is scalable if there are a few agents. However, when considering thousands of agents and a central ILP which requires values for all different joint action pairs, there is a combinatorial explosion making the model non deployable in real time. To put things into perspective, assuming each agent has $|A|$ feasible actions (request combinations) and there are $N$(typically 1000) agents, the number of value evaluations jumps from $N \cdot |A|$ in **DJV** to $N \cdot |A|^2$ while using a coordination graph. It should be further noted the $A$ corresponds to request combinations and hence can increase combinatorially.

Thus, we need a mechanism that is scalable while considering the impact of $i$ on other agents. In the well known Expectation Maximization algorithm [Dempster (1977)] for identifying missing data, the likelihood is calculated by introducing a conditional probability of unknown data given known data. In a similar vein, our method to deal with the unknown impact of other agents is by considering conditional probability of agent $j$ taking action $a_j$ given agent $i$ takes action $a_i$ in state $s$. This will ensure the overall value is dependent on individual agent values and not on joint values. More specifically, the expected value of agent $i$ is:

$$Q_i^{CG}(s,a) = f^i(a^i|s) + \sum_{j|(i,j) \in E, a_j \in A_j} P(a_j|a_i, s) f^j(a^j|s)$$

To make this broad idea of conditional expectation operational in case of RMP, we have to address multiple key challenges. We describe these key challenges and our ways of addressing them below. Figure 2 provides the overall method, with step (II) outlining the conditional expectation idea and the key difference from NeurADP described in Figure 1.

### 3.1 NO EXPLICIT/STATIC COORDINATION GRAPH

While it is clear that agents that are very far apart will not have any dependency, there is no explicit coordination graph that is present in RMP. However, RMP has two characteristics that make it easier to identify neighbouring agents for any given agent:

- Agents that are nearby spatially are more probable to compete over the same set of requests and hence would have a dependency.

- Agents/vehicles do not have identity, i.e., they are all homogenous.

Due to these characteristics, we can cluster the intersections in the road network (to capture spatial dependencies) and consider agents at a time step in an intersection cluster as neighboring agents. Due to homogeneity of agents, the only aspect of importance is whether there are agents (and not which specific agents) competing for the same requests. Unlike previous works, the coordination graph keeps changing at each time step, as agents move between clusters. At time $t$, an agent placed at an intersection belonging to cluster $C_k$ will coordinate with all other agents in cluster $C_k$.

We define function $\mathcal{M} : \mathcal{L} \longrightarrow [K]$ to map each intersection of the road network into one of the K clusters : $\{C_1, C_2, \ldots, C_K\}$ based on the average travel times between intersections. Because of clustering locations and assigning agents to location clusters, the total number of agents becomes less of an issue with respect to scalability.

---

[2]We will use agent and vehicle interchangeably.

### 3.2 How to consider impact of other agents in the cluster?

Let us consider agent $i$ present in cluster $C_k$ at time $t$. The other agents in cluster $C_k$ are agents $j_1, j_2, \ldots, j_n$ and are termed its neigbours. In NeurADP, agent $i$ credited action $f \in \mathcal{F}_t^i$ with value $V^i(s_t^{i,f})$ which is oblivious to the presence of neighbour agents. Since the execution is centralized, agent $i$ can however weigh the losses/gains its action $f$ has on the cluster by getting useful feedback from the individual values of other agents. Let us take a neighbour agent $j \in C_k$ $(j \neq i)$ having feasible actions $g_1, g_2, \ldots, g_{k_j} \in \mathcal{F}_t^j$. From agent $i$'s perspective a conditional probability distribution

$$P(\text{Agent j takes action g}|\text{Agent i takes action f}) = P^j(g|s_t^i, f)$$

is formed and the feedback term from agent $j$ is written as $\sum_{g \in \mathcal{F}_j^t} P^j(g|s_t^i, f)V^j(s_t^{j,g})$.

We do this for all the neighbours and take an average :

$$\frac{1}{|C_k| - 1} \sum_{j \in C_k, j \neq i} \sum_{g \in \mathcal{F}_j^t} P^j(g|s_t^i, f)V^j(s_t^{j,g})$$

Now to calculate the value of agent $i$ on taking action $f$, after getting this feedback, we take an affine combination and write the new individual value as

$$\hat{V}^i(s_t^{i,f}) = \frac{1}{1 + \lambda}\left[V(s_t^{i,f}) + \lambda \frac{1}{|C_k| - 1} \sum_{j \in C_k, j \neq i} \sum_{g \in \mathcal{F}_j^t} P^j(g|s_t^i, f)V^j(s_t^{j,g})\right]$$

where $\lambda$ is a learnable parameter. ***It should be noted that this individual value not only considers the future impact of current action, $f$, but also considers the impact on other agents.***

Our overall ILP objective thus becomes :

$$\max \sum_i \sum_{f \in \mathcal{F}_i^t} \left(\mathcal{J}^i(s_t^i, f) + \gamma \frac{1}{\lambda + 1}\left[V^i(s_t^{i,f}) + \lambda \sum_{j \in C_k, j \neq i} \frac{1}{|C_k| - 1} \sum_{g \in \mathcal{F}_j^t} P^j(g|s_t^i, f)V^j(s_t^{j,g})\right]\right) \times a_t^{i,f}$$

$$= \max \sum_i \sum_{f \in \mathcal{F}_i^t} \left(\mathcal{J}^i(s_t^i, f) + \gamma \hat{V}^i(s_t^{i,f})\right) \times a_t^{i,f}$$

subject to feasibility constraints in expression 2.

**What should be the functional form for conditional probabilities?** Each request $u_t^i \in \mathcal{U}_t$ has a pickup location $o_t^i \in \mathcal{L}$. Recall function $\mathcal{M} : \mathcal{L} \longrightarrow [K]$ which maps each intersection to its cluster. Every action $f$ is associated with some user request $u_t^i$, we define

$$\mathcal{A}_t : \mathcal{F}_t \longrightarrow \mathcal{L}$$

which maps each action $f$ to the corresponding request $(u_t^i)$'s pickup intersection $o_t^i$. Define the composition function

$$\mathcal{C}_t = \mathcal{M} \circ \mathcal{U}_t : \mathcal{F}_t \longrightarrow [K]$$

that maps each action to a cluster by the pickup location of the action's corresponding user request. $d : [K] \times [K] \longrightarrow \mathcal{R}^+$ is defined as the average travel time between 2 clusters. We model the conditional probability of agent $j$ taking action $g$ given agent $i$ takes action $f$ as :

$$P^j(g|s_t^i, f) \propto e^{(\alpha \cdot d(\mathcal{C}_t(g), \mathcal{C}_t(f)))}$$

where $\alpha$ is a learnable parameter. The normalizing constant is computed by summing over actions in $\mathcal{F}_t^j$.

### 3.3 Over/under estimation of individual values in **FV** and **DJV**:

NeurADP makes an optimistic assumption that individual agent will get to take the best action in the next time step. Since central ILP decides the joint action, this can result in an overestimation or underestimation of individual agent value. Due to this and other issues, in our experiments, we found that NeurADP values can have significant errors compared to the discounted future rewards as shown in Figure 3. We fix this problem through two key enhancements:

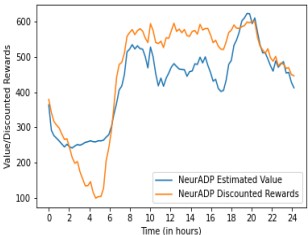 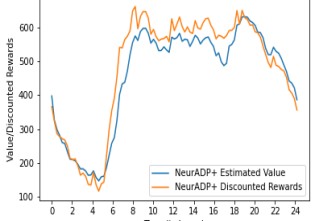 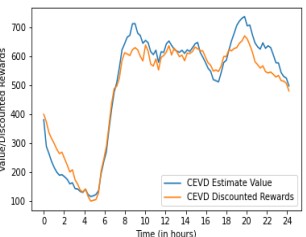

Figure 3: Comparison of estimated value and discounted "real" value for NeurADP, NeurADP+ and CEVD.

- Controlling large Variance of exogeneous information : Recall from Equation 4, to calculate values of post action states, NeurADP employs

$$V^a(s_t^a) = \mathbb{E}[V(s_{t+1})|s_t^a, \xi_{t+1}]$$

Here the exogenous information $\xi_{t+1}$ is the global demand ($g_t$) at time $t$. Even within a small number of consecutive epochs, the global demand displays a significant variance. This results in individual values showing a large variance, instead of varying smoothly over time. We thus consider expected discounted future demand,

$$F_{t+1} = \mathbb{E}[\sum_{t'=0}^{T} \gamma^{t'} g_{t+t'}]$$

(where $T$ is large but finite horizon) which varies smoothly over time unlike the current demand. In our approach, we consider exogenous information as $\xi_{t+1} = [g_t, F_t]$.

- Enforcing values to be positive : NeurADP models the value function as a shared parameter Neural Network. The final layer of this Neural Network is a fully connected multi layer perceptron (MLP) having range as the entire real line. However, as our objective function (number of requests served) is non-negative, negative values (admissible by NeurADP) are not reasonable. We thus use a SoftPlus activation after the final MLP to ensure that the computed values are strictly positive.

  To evaluate the impact of the above modifications (calling the model NeurADP+), in Figure 3 we plot the following : $\sum_i V^i(s_t^{f_t^i, i})$ ($f_t^i$ is the action chosen for vehicle $i$ at time $t$) vs $\sum_{t'=0}^{T} \gamma^{t'} R_{t+t'}$ where $R_t$ is the total number of requests served at time $t$ for NeurADP, NeurADP+ and CEVD. Notice how the gap between the Estimated Value curve and the Discounted Reward curve is very small in NeurADP+ and CEVD (as it should be by the Bellman Equation 3), whereas the gap is quite significant for NeurADP. Note that the height of the graphs is different and while NeurADP+ improves the quality of the estimation, CEVD is responsible for the bulk of the performance gain.

## 4 ALGORITHM

Given a post decision state $s_t^{i,f}$, we need a paremterized function to compute $\hat{V}(s_t^{i,f})$. Our joint function $\hat{V}_{\theta,\lambda,\alpha}$ has 3 parameters : (i) $\theta$ : parameters of a Neural Network Based Individual Agent Value Function Estimator (as in NeurADP)[3], (ii) $\lambda$ : parameter to control the importance given to an agent's neighbours while taking an affine combination, (iii) $\alpha$ : parameter to control conditional probabilities **P** which controls the relative importance given to different feasible actions of neighbours. We infer the parameters step by step. Setting $\lambda = 0$, reduces this function to NeurADP ($\hat{V}_{\theta,0,0}$). We first estimate optimal $\theta^*$ (following the alogrithm in NeurADP). This gives us the NeurADP parameters, which are a good starting point to estimate values at an individual level. Now to estimate $\lambda$, we set $\alpha = 0$ (this corresponds to uniform distribution over actions), and do a linear search on a set of sampled points on the real

---

[3]In this section by NeurADP we mean NeurADP updated with the proposals in 3.3

line to find the optimal $\lambda^* = \max_\lambda \sum_{t=0}^{t=\mathcal{T}} \sum_{i=0}^{|R|} \left( \mathcal{J}^i(s_t^i, f) + \gamma \hat{V}(s_t^{i,f}) \right) \times a_t^{i,f}$ subject to constraints in expression 2. At this stage, we haven't changed the preference over actions from an individual perspective, however for the central executor the values are now much more refined as the individual over/under estimates have been smoothened by considering the neigbours. Finally we estimate $\alpha$ by linear search on a set of sampled points on the real line to get optimal $\alpha^* = \max_\alpha \sum_{t=0}^{t=\mathcal{T}} \sum_{i=0}^{|R|} \left( \mathcal{J}^i(s_t^i, f) + \gamma \hat{V}(s_t^{i,f}) \right) \times a_t^{i,f}$ subject to constraints in expression 2. We can now compute values on unseen data using $\hat{V}_{\theta^*, \lambda^*, \alpha^*}$.

## 5 EXPERIMENTS

The goal of the experiments is to compare the performance of our approach CEVD to NeurADP[Shah et al. (2020)](henceforth referred to as baseline), which is the current best approach for solving the RMP. We make this comparison on a real-world dataset [NYYellowTaxi (2016)] across different RMP parameter settings. We quantitatively justify our performance by comparing the service rate, i.e., the percentage improvement on the total requests served. We vary the following parameters: the maximum allowed waiting time $\tau$ from 90 seconds to 150 seconds, the number of vehicles $|\mathcal{R}|$ from 500 to 1000 and the capacity $C$ from 4 to 5. The value of maximum allowable detour delay $\lambda$ is taken as $2 * \tau$. The decision epoch duration $\Delta$ is taken as 60 seconds.

**Setup:** We perform our experiments on the demand distribution from the publicly available New York Yellow Taxi Dataset [NYYellowTaxi (2016)]. The experimental setup is similar to the setup used by [Shah et al. (2020)]. Street intersections are used as the set of locations $\mathcal{L}$. They are identified by taking the street network of the city from openstreetmap using osmnx with 'drive' network type [Boeing (2017)]. Nodes that do not have outgoing edges are removed, i.e., we take the largest strongly connected component of the network. The resulting network has 4373 locations (street intersections) and 9540 edges. The travel time on each road segment of the street network is taken as the daily mean travel time estimate computed using the method proposed in [Santi et al. (2014)]. We further cluster these intersections $\mathcal{L}$ into $K$ clusters using K-Means Clustering based on the average travel times between different intersections. We choose the value of $K$ based on the number of vehicles. $K$ is chosen to be 100,150 and 200 for 500,750 and 1000 vehicles respectively. Similar to previous work, we only consider the street network of Manhattan as a majority ($\sim 75\%$) of requests have both pickup and drop-off locations within it. The dataset contains data about past customer requests for taxis at different times of the day and different days of the week. From this dataset, we take the following fields: (1) Pickup and drop-off locations (latitude and longitude coordinates) - These locations are mapped to the nearest street intersection. (2) Pickup time - This time is converted to appropriate decision epoch based on the value of $\Delta$. The dataset contains on an average 322714 requests in a day (on weekdays) and 19820 requests during peak hour.

We evaluate the approaches over 24 hours on different days starting at midnight and take the average value over 5 weekdays (4 - 8 April 2016) by running them with a single instance of initial random location of taxis [4]. CEVD is trained using the data for 8 weekdays (23 March - 1 April 2016) and it is validated on 22 March 2016. For the experimental analysis, we consider that all vehicles have identical capacities.

**Results :** We compare CEVD to NeurADP (referred to as baseline). Table 1 gives a detailed performance analysis for the service rates of CEVD and baseline. Here are some key observations:
**Effect of changing tolerance to delay, $\tau$:** CEVD obtains a 9.37% improvement over the baseline approach for $\tau = 90$ seconds. The difference between the baseline and CEVD decreases as $\tau$ increases. The lower value of $\tau$ makes it difficult for vehicles to accept new requests while satisfying the constraints for already accepted requests. The neighbouring vehicles' interactions in CEVD prevents a vehicle from picking up requests it values highly however which would have been more suitable given the delay constraints for some other vehicle and instead picks up requests which it might value less but still is feasible. Thus the overall requests served increases.
**Effect of changing the capacity, $C$ :** CEVD obtains a 9.76% gain over baseline for capacity 5. The

---

[4]All experiments are run on 60 core - 3.8GHz Intel Xeon C2 processor and 240GB RAM. The algorithms are implemented in python and optimisation models are solved using CPLEX 20.1

| Varying | Parameters | | | Baseline | Our Approach | |
| --- | --- | --- | --- | --- | --- | --- |
| | Number of Vehicles | Pickup Delay | Capacity | Requests Served | Requests Served | Percentage Improvement |
| Pickup Delay | 500 | 90 | 4 | 90286.8±2108.78 | 98748.2±2449.38 | 9.37±0.59 |
| | 500 | 120 | 4 | 103933.0±2604.05 | 113184.0±2774.00 | 8.90±0.23 |
| | 500 | 150 | 4 | 113051.8±2771.45 | 117351.6±2818.49 | 3.80±0.24 |
| Number of Vehicles | 500 | 90 | 4 | 90286.8±2108.78 | 98748.2±2449.38 | 9.37±0.59 |
| | 750 | 90 | 4 | 129791.6±3516.83 | 139365.6±3853.98 | 7.38±0.38 |
| | 1000 | 90 | 4 | 165110.2±4677.10 | 175453.4±5173.82 | 6.26±0.18 |
| Capacity | 500 | 90 | 4 | 90286.8±2108.78 | 98748.2±2449.38 | 9.37±0.59 |
| | 500 | 90 | 5 | 91509.4±2144.54 | 100443.8±2502.34 | 9.76±0.38 |

Table 1: Detailed Quantitative Results

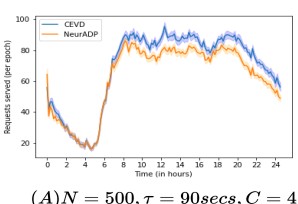
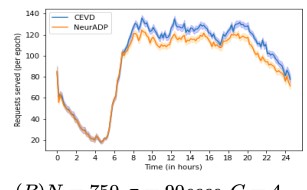
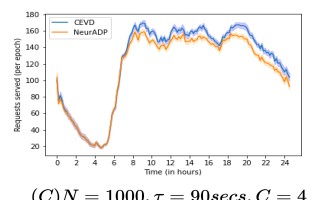

$(A) N = 500, \tau = 90 secs, C = 4$    $(B) N = 750, \tau = 90 secs, C = 4$    $(C) N = 1000, \tau = 90 secs, C = 4$

Figure 4: This graph compares the number of requests served as a function of time. The bold lines represent a moving average for different configurations averaged over requests from 4-8 April 2016.

difference between the baseline and CEVD increases as the capacity increases as for higher capacity vehicles, there is a larger scope for improvement if vehicles cooperate well.

**Effect of changing the number of vehicles, $|\mathcal{R}|$:** CEVD obtains a 9.37% improvement over the baseline for capacity $|\mathcal{R}| = 500$. The difference between the baseline and CEVD decreases as the number of vehicles increase as in the presence of a large number of vehicles, there will always be a vehicle that can serve the request. As a result, the quality of assignments plays a smaller role.

We further analyse the improvements obtained by CEVD over baseline by comparing the number of requests served by both approaches at each decision epoch throughout the day. Figure 4 shows the number of requests served by the baseline and CEVD at different decision epochs[5]. As shown in the figure, initially at night time when the demand is low both approaches serve all available demand. During the transition period from low demand to high demand period, the baseline algorithm starts to choose suboptimal actions without considering the impact of each vehicle's action on the whole system while CEVD is able to capitalize on the joint action values and serve much more requests than the baseline.

The approach can be executed in real-time settings. The average time taken to compute each batch assignment using CEVD is less than 60 seconds (for all cases) [6]. These results indicate that using our approach can help ride-pooling platforms to better meet customer demand.

## 6 CONCLUSION

Due to the matching required on a tri-partite graph between user requests, trips (combination of user requests) and vehicles, on-demand ride pooling is challenging. Improving on existing methods, we provide a scalable novel value decomposition method based on conditional probabilities, where individual value is not only able to consider future impact of current matches but also impact on other agent values. This new approach is able to outperform the best existing method in all settings of the benchmark taxi data set employed for on-demand ride pooling by margins of up to 9.79%. To put this result in perspective, typically, an improvement of 1% is considered a significant improvement on ToD for an entire city [Xu et al. (2018); Lowalekar et al. (2019)].

---

[5]Results for other settings shown in appendix

[6]60 seconds is the decision epoch duration considered in the experiments

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
