# OpenReview forum: "Conditional Expectation based Value Decomposition for Scalable On-Demand Ride Pooling"
_ICLR.cc/2022/Conference — ICLR 2022 Submitted_

### Official Review · Reviewer_9FHL · 2021-11-01

**Correctness:** 3
**Technical Novelty And Significance:** 2
**Empirical Novelty And Significance:** 3
**Recommendation:** 5
**Confidence:** 4

**Main Review:**

# Pros and Cons

## Pros
- The performance gain achieved by the paper is quite significant from the application perspective.
- The idea of considering the effect of other agents seems to be correct and valuable for multi-agent decision-making problems like ToD systems.
- The computational efficiency is shown (i.e., the optimization is done within the batch time of 60 seconds).

## Cons

- Some notations are not clearly defined, and it is hard to follow the details of the CEVD and its idea.

# Comments

To clearly understand the paper and clarify its score, I would like to clarify the following point.

(I) Many notations in Sec. 2 are not explained or defined. These notations or missing explanations make readers hard to follow the proposed study and know the difference between NeurADP and CEVD.

The below are examples.

(I-1) Eq.(1): What is $[\mathcal{U}]^{c'}$?

(I-2) Between Eq.(2) and Eq.(3): What are $T^a(\cdot,\cdot)$ and $T^\xi(\cdot, \cdot)$?

(I-3) Page 4: What is $V^i(\big\langle r_t^{i,a}, r_t^{-i,a}\big\rangle)$? I cannot follow this $\langle\cdot\rangle$ notation.

(I-4) Page 4: What is the definition of $r_t^a$?

(I-5) Page 5 before sec 3.1: $\sum_{j\in E}$ means $\sum_{j|(i, j)\in E}$? What is the difference between Pr and P?: $\mathit{Pr}(a_j\mid a_i, s)$ and $\mathit{P}(\text{Agent j takes action g | Agent i takes action f})$.

(I-6) Page 5: What is the definition of $P^j$. Agent $j$'s probability?

(I-7) Page 6: What is the definition of $s_t^{i,f}$?

(I-8) Page 7: What is the definition of this bracket $[g_t, F_t]$ of $\xi_{t+1}$?

(II) What are 'other issues' of the over/under estimation of individual values in FV and DJV? (Page 6). Please give an example of such issues.

(III) In optimization (sec 4), the proposed method estimates $\theta, \lambda, \alpha$ step by step. Please clarify the property of this optimization problem. Are the resulted $\lambda^\star,\alpha^\star,\lambda^\star$ are globally optimal?

(IV) Please clarify the relation between experimental results with the size $N$. That is, experimental results are reported only with $N=500, 750, 1000$. However, some research have modeled the ToD service with more agents (e.g., 3000 agents in [Alonso-Mora et al. 2017]), and therefore, I'm interested in more details of this aspect.

For example, the authors reported that "The average time taken to compute each batch assignment using CEVD is less than 60 seconds (for all cases)". Is this due to the size $N\leq 1000$ or not?

Further, are computational results affected by the clustering ($k$-means) or not?

(V) Please clarify the overview of the computation of CEVD (e.g., with a pseudo-code or system overview). What are the input and output? How is the ILP solver (CPLEX) used? Which part consumed the time within 60 seconds? I guess that such additional information is helpful for readers.

**Summary Of The Paper:**

This paper studies an RPM problem (ride-pool matching problem) for on-demand transportation services. This problem is recently studied in various papers, but it is hard to choose a good matching by just using a bipartite graph matching due to the future demands, and it is an online decision-making problem.

A recent breakthrough, NeurADP by [Shah et al. 2020], has shown a good performance, but the proposed approach in the paper, CEVD, achieved much more performance gain (reported as 3.8%-9.76%), which has a significant impact on the ToD service. An essential technique of the proposed CEVD is considering the effect of other agents (i.e., other vehicles) when estimating the value of actions.

**Summary Of The Review:**

To the best of my knowledge and experiences, the achieved performance of CEVD seems to show important progress on the ToD services. The computational times and settings are reasonable.

However, the paper contains some unclarified notations. It is hard to follow the details of the proposed method. I want the authors to give more explanations and clarify some parts to help readers, which is also important to increase my review score.

---

> ### Author Response · Authors · 2021-11-18
> **Thank you for the review**
>
> ***I***
>
> We apologize for some of the notational inconsistencies which occurred while going on from related work to our method. We have updated these changes. We hope to clarify your doubts below and have added a note in the main paper for readers to refer Appendix A.1 (can be found in updated supplementary zip) for a detailed description of the RMP notations.
>
> (I-1) $[\mathcal{U}]^{c'}$ is the \textbf{$c'$-ary Cartesian power} of the set $\mathcal{U}$ (set of requests). An action assigned to a taxi can have $c'$ distinct user requests, where $c'$ is less than or equal to $c^i$ - the maximum capacity of the agent. Thus, the action space is the union of varying number of requests, i.e., the \textbf{$c'$-ary Cartesian power} of $\mathcal{U}$, where $c'$ goes from 1 to $c^i$.
>
> (I-2) Recap that $s_t$ is the current state, $a_t$ is the chosen action, $\xi_{t+1}$ is the exogenous information (e.g. position of other taxis, requests and trips) at time $t+1$, $s_t^a$ is the post action state from $s_t$ upon taking action $a_t$ and $s_{t+1}$ is the state after the vehicles have moved some distance and new requests have arrived (concisely captured in $\xi_{t+1}$). ${T}^a(s_t,a_t)$ and ${T}^\xi(s_t^a,\xi_{t+1})$ are the transition functions. We denote then by ${T}^a(.,.)$ and ${T}^\xi(.,.)$ to talk of them as functions taking 2 inputs. ${T}^a(.,.)$ gives the post action state $s_t^a$ upon taking inputs $s_t$ and $a_t$. ${T}^\xi(.,.)$ returns $s_{t+1}$ upon taking inputs $s_t^a$ and $\xi_{t+1}$.
>
> (I-3) We had used both $<,>$ and $[,]$ to denote concatenation. We have now made all concatenations consistent to $[,]$ in the and also added a footnote to clear ambiguity regarding this non-standard notation.
>
> (I-4) $r_t^a$ (notation used in NeurADP) is the same as post state action $s_t^a$ described above. We have changed it to $s_t^a$ everywhere.
>
> (I-5,6) Thanks for pointing out. $j \in E$, will indeed be $j|(i,j) \in E$. Also Pr() and P() both denote the same conditional probability as stated, we have fixed it to P() to avoid ambiguity.
>
> (I-7) $s_t^{i,f}$ is the post action state for agent $i$ upon taking action $f$. This is in similar vein as $s_t^a$ describe above, but defined specifically for agent $i$.
>
> (I-8) $[g_t, F_t]$ is simply the concatenation of current global demand : $g_t$ and expected discount future demand : $F_t$.
> \end{itemize}
>
> ***II***
>
> Others issues are: (i) Large variance of exogenous information, i.e., demand; (ii) Negative values can be output from the Q network. Our approaches presented in the section address these two issues.
>
> ***III***
>
> The algorithm used to estimate our model parameters is based on Expectation Maximization, which is guaranteed to reach a local optimum.
>
> ***IV***
>
> 3000 agents: We present the results for $N=3000$ in ***Computational Time*** (in Common Response above).  This shows that our method performs well even in settings with a large number of agents. Please note that with so many agents, prior work already achieves excellent performance. The advantage with our new method is that we can achieve similar performance with fewer agents.
>
> 60 seconds:  Note that even with 3000 agents, the time taken to compute matching decisions is at the maximum only 3.5 seconds, which is well below 60 seconds.
>
> k-means clustering: Clustering does have some impact as described in ***Computational Time*** (in Common Response above) and the corresponding Table , but CEVD remains executable in real time settings.
>
> ***V***
>
> Please refer to ***Pseudocode*** (in Common Response above). We thank you for pointing out that a pseudocode will greatly aid the overall understanding of our algorithm. We have also incorporated this in Appendix A.3 (in updated supplementary zip).
>
> We hope this clarifies the reviewer's concerns on the notations. Let us know if you have any other questions.

---

> > ### Comment · Reviewer_9FHL · 2021-11-24
> > **Thank you for your updates**
> >
> > Dear authors,
> >
> > Thank you for your updates, I'll carefully check the comments, manuscript, and appendix.

---

### Official Review · Reviewer_kKMH · 2021-11-02

**Correctness:** 3
**Technical Novelty And Significance:** 3
**Empirical Novelty And Significance:** 3
**Recommendation:** 8
**Confidence:** 3

**Main Review:**

The proposed approach in the paper considers the impact of other agents actions on individual value by computing conditional expectations to improve the overall performance. The experimental results verify that the CEVD is an effective method for improving the overall requests served by 9.76 compared to the baseline, which seems promising. This work is quite innovative and the paper is generally well-written. Some concerns are shown as follows:

-- The background of the abstract is a little longer.
-- The first paragraph should be the definition of Approximate Dynamic Program (ADP) rather than FV because the definition of FV was given in the previous paragraph.
-- The mathematical symbols of this paper are a bit too many, and it is recommended to list a table to show the meaning of each symbol to facilitate the reader to understand the paper.
-- I am very confused about the data features used in K-Means to cluster intersections to clusters. As stated in the last paragraph of page 5, the function M is clustering locations. But in the second paragraph of section 5 experiments, it says that K-Means clustering is based on the average travel times between different intersections. So which feature is used? locations or average travel times? This needs to be explained clearly and it is crucial to experiment setups.
-- In Section 6 conclusion, I highly recommend to the authors to add more open issues and future directions of their work.


**Summary Of The Paper:**

This paper focuses on the ride-pool matching problem to efficiently allocate combinations of user requests to vehicles online under quality constraints and matching constraints. Intending at this, the authors come up with a conditional expectation based value decomposition (CEVD) method.

**Summary Of The Review:**

In general, the paper makes very solid work and is suited to be published in ICRL.

---

> ### Author Response · Authors · 2021-11-18
> **Thank you for the review**
>
> We thank the reviewer for the positive review appreciating our contribution and results.
>
> ***Data features used to create clusters:***
>
> The road network ${\cal G}$ has intersections : ${\cal L}$ as nodes, road segments :  ${\cal E}$ as edges and weights on edges indicate the travel time on the road segment. We represent each node, $i$ using a $|{\cal L}|$ dimensional vector, $\Gamma_i$ that represents the average time from that node to all other nodes. For the $i^{th}$ node with vector $\Gamma_i$,  the $j^{th}$ value in the vector, i.e., $\Gamma_{i,j}$, represents the average time from $i^{th}$ node to $j^{th}$ node. Also, $\Gamma_{i,i} = 0$, as the time between a node and itself is zero. We use K-means clustering on these vectors $\{\Gamma_1, \ldots, \Gamma_{|\mathcal{L}|}\}$.

---

### Official Review · Reviewer_Siyp · 2021-11-03

**Correctness:** 3
**Technical Novelty And Significance:** 2
**Empirical Novelty And Significance:** 2
**Recommendation:** 5
**Confidence:** 4

**Main Review:**

Strengths
- Ridesharing (pooling) is a challenging domain for RL/ADP. The paper presents an incremental step forward on top of NeurADP for the matching problem.
- The paper proposes a way to incorporate agent interactions with neighbors without relying on joint action values, which suffer from the curse of dimensionality.
- Different values of problem configuration parameters are investigated in the experiments.
- The experiment dataset is public so that reproducing the results is possible.


Weaknesses and comments
- The paper uses the number of completed requests as the problem objective. In practice, a common measure for ride-pooling is the ratio between the sum of completed individual trip distances (as if they were fulfilled as single trips) and the total distance that the drivers actually travel to fulfill those requests. This metric measures both the amount of request fulfilled and the quality of the pooling.

- The proposed estimation of agent interaction effect involves a handcrafted conditional agent probability based on a softmax over distance between destination pairs. There's no empirical justification on this choice. Why should the probability based on how far away are the action destinations? And, matching decision is ultimately a system decision, so shouldn't it be learned from data of system decisions?

- Figure 3: This may not be the impact of enforcing positive values. This is just showing that CEVD learns the individual values better than NeurADP. It could be because of its consideration of impact on neighboring agents. I suggest the authors do a more careful ablation study to separate the effect of multiple algorithmic differences.

- Section 4: For finding optimal \lambda, I don't see \lambda appearing here in the expression. How did you tune it exactly?

- The paper claims that the algorithm can be executed in real-time setting. However, the decision epoch in practice is much shorter, only a few seconds. A run time of 60 seconds to compute a matching decision is way too slow for 'real-time execution'.

- There are too many mentions of "e.g., Uber, Lyft, and Grab". Once is good enough, and we all know they are well-known ridesharing companies.

**Summary Of The Paper:**

This paper considers the ridesharing matching problem and builds the solution upon NeurADP. The main contribution over NeurADP is that the action values of each agent (vehicle) takes into account the impact of its action on the neighboring agents within the same cluster, which is obtained through clustering of the intersections on the road network. The impact of agent's action on neighbors is measured by the neighboring agents' independent values weighted by their action probabilities conditional on the agent's action. Benchmarking was performed on the NYC taxi data set against NeurADP. Results on different values of tolerance for delay, capacity, and number of vehicles are reported, and a significant improvement is demonstrated in all cases.

**Summary Of The Review:**

The paper improves upon NeurADP by proposing a way to incorporate agent interactions without resorting to joint action values. While there's some merit in technical contribution in this regard, there are a number of major issues in algorithmic justification and empirical validation.

---

> ### Author Response · Authors · 2021-11-18
> **Thank you for the review**
>
> ***Objective:*** Our modelling approach doesn't restrict the choice of objective function $\mathcal{J}(s_t, a_t)$ and can easily be extended to the proposed objective.
>
> To ensure fair comparison with the existing best approach, NeurADP and following the objective employed in existing papers [1,2,3], we used number of completed requests as the problem objective.
>
> ***Justification for handcrafted conditional probability using Softmax on action distances:***
>
> If we were to consider an arbitrary functional form (potentially a neural network) to estimate the conditional probabilities, it would require gradients to propagate through a Mixed Integer Program, which is incredibly challenging in the problems of interest where MIP can have thousands of constraints. If the central agent had a small linear optimization problem, then we could potentially have utilized end to end learning to obtain the conditional probabilities.
>
> We tried other deterministic functional forms like dot products of post action states, difference of average demand in action destinations but they were unable to capture beneficial conditional probabilities. The current Softmax form is a low overhead choice that works incredibly well, improving by upto $9.76$% over the state of the art.
>
> Softmax itself is not an unexpected choice, as it has been used when probabilities need to be computed based on feature values -- exploration in multi-armed bandits based on average value, Quantal response strategies based on value function, classification problems with cross entropy loss based on scores for different categories etc.
>
> Prior work such as [4] use multi dimensional feature representation to compute attention weights between a pair of agents to measure their strength of connection. They particularly highlight the importance of inter-agent distance and corresponding actions.
>
> Furthermore, interactions between agents (vehicles) in ride pooling setting occur because of trying to service common demand. The reason for considering action destinations in deriving conditional probability values is because if agents are near to each other after serving requests, then there is a higher chance of an interaction between the vehicles after serving their current requests.
>
> Thus (i)The lack of gradients through MIP making end to end training very hard, (ii)Softmax being a popular choice to represent probabilities, (iii As stated in ***computational time*** (common response) this expression also comes at very little computational overhead as we leverage pre-computed values, (iv)Prior work stressing on inter-agent distance as an important factor despite availability of multi-dimensional features, serves as the motivation behind our choice of conditional probability representation based on action distances.
>
> ***Figure 3*** We apologize for the confusing positioning of the sentence. We should have indicated the figure in a separate sub-section. The reviewer has rightly pointed out that we directly moved from NeurADP to CEVD, without providing the implication of the two modifications introduced in the section. We updated Figure 3, which now provides the ablation study with both the modifications to NeurADP,termed NeurADP+ which shows that individual values learned are much better than in NeurADP. Note that the heights of the three graphs are different and the bulk of the performance gain is through CEVD.
> ***Section 4: How to find optimal $\lambda$***
>
> We request the reviewer to please refer to section 3.2 below the sentence starting with "Now to calculate the value of agent i on taking action f...", where the expression for $\hat{V}(s_t^{i,f})$ is defined.
>
> Expanding the expression becomes clumsy, so we used it in the the abstracted form in section 4. In Section 4 itself, we also mention $\hat{V}$ has 3 parameters, and the exact role of $\lambda$. For the estimation of $\lambda$, we refer the reviewer to Appendix A.6 (in updated supplementary zip).
>
> ***Decision epoch; Time to compute matching decision***
> There seems to be a misunderstanding. Please refer to the description in Section 2 (RMP) : "Passengers that wish to travel from one location to another send requests to a central entity that collects these requests over a time-window called the decision epoch $\Delta$". Thus, decision epoch is the time over which requests are accumulated. The actual time taken to compute a matching decision (i.e., computing action feasibility sets, computing values and then solving the MIP) is a few seconds as represented in the table in the ***Computational Time***(Common Response).
>
> Also, a decision epoch of 60secs is standard in  literature [1,2,3].
>
> [1] Alonso Mora et al, On-demand high-capacity ride-sharing via dynamic trip-vehicle assignment, Proceedings of the National Academy of Sciences, 2017
>
> [2] Shah et al, NeurADP, AAAI, 2020
>
> [3] Lowalenkar et al, ZAC, JAIR, 2021
>
> [4] Li et al, Deep Implicit Coordination Graph for multi-agent reinforcement learning, AAMAS, 2021

---

### Author Response · Authors · 2021-11-18
**Common Response to all Reviewers (1/2)**

We thank all the reviewers for their valuable and constructive feedback. We appreciate that you find our approach novel and our performance gain beneficial for applications.

***Pseudocode:*** We present below the pseudocode for CEVD Execution.

1. ***Input :*** Neural Network Value Function $V$, parameters $\lambda$ (controlling relative importance of neighbours) and $\alpha$ (controlling conditional probabilities), Number of Clusters : K, Road Network : $\mathcal{G}$, Decision Epoch : $\Delta$, Delay Parameter : $\tau$, Capacity : $C$, Objective Function : $\mathcal{J}$
2. Compute K clusters on road network $\mathcal{G}$ based on average travel times between intersections.
3. Precompute conditional probabilities (upto a proportionality constant) and store in a K $\times$ K matrix.
4. Initialize the state $s_{0}$ by randomly positioning vehicles.
5. for each step $0 \leq t \leq T$
6. &nbsp;&nbsp;&nbsp;&nbsp;&nbsp;&nbsp; Fetch all user requests $\mathcal{U}_t$ in decision epoch $\Delta$
7.&nbsp;&nbsp;&nbsp;&nbsp;&nbsp;&nbsp; Compute the feasible action set $\mathcal{F}_t$ based on current state $s_t$ and the user requests.
$\mathcal{F}_t^i$ =

&nbsp;&nbsp;&nbsp;&nbsp;&nbsp;&nbsp;&nbsp;&nbsp;&nbsp;&nbsp;&nbsp;&nbsp;&nbsp;&nbsp;&nbsp;&nbsp;&nbsp;&nbsp;$\{f^i | f^i \in \cup_{c'=1}^{C} [\mathcal{U}]^{c'}, \text{PickUpDelay}(f^i,i) \leq \tau, \text{DetourDelay}(f^i,i) \leq 2 * \tau \}$

8. &nbsp;&nbsp;&nbsp;&nbsp;&nbsp;&nbsp;Compute individual values using the Neural Network Value Function :

&nbsp;&nbsp;&nbsp;&nbsp;&nbsp;&nbsp;&nbsp;&nbsp;&nbsp;&nbsp;&nbsp;&nbsp;&nbsp;&nbsp;&nbsp;&nbsp;&nbsp;&nbsp;$V^i(s_t^f);\forall f \in \mathcal{F}_t^i, \forall i \in {\cal R}$

9. &nbsp;&nbsp;&nbsp;&nbsp;&nbsp;&nbsp;Assign each agent its cluster based on current location.
10. &nbsp;&nbsp;&nbsp;&nbsp;&nbsp;&nbsp;Compute CEVD Values $\forall f \in \mathcal{F}_t^i, \forall i \in {\cal R}, \forall k \in [\text{K}]$ :

&nbsp;&nbsp;&nbsp;&nbsp;&nbsp;&nbsp;&nbsp;&nbsp;&nbsp;&nbsp;&nbsp;&nbsp;&nbsp;&nbsp;&nbsp;&nbsp;&nbsp;&nbsp; $\hat{V}^i(s_t^{i,f}) = \frac{1}{1 + \lambda}\bigg[V(s_t^{i,f}) + \lambda \frac{1}{|C_k| - 1} \sum_{j \in C_k, j \neq i} \sum_{g \in \mathcal{F}^t_j} P^j({g|s^i_t,f})V^j(s_t^{j,g})\bigg]$

11. &nbsp;&nbsp;&nbsp;&nbsp;&nbsp;&nbsp; Solve the MIP :

&nbsp;&nbsp;&nbsp;&nbsp;&nbsp;&nbsp;&nbsp;&nbsp;&nbsp;&nbsp;&nbsp;&nbsp;&nbsp;&nbsp;&nbsp;&nbsp;&nbsp;&nbsp;$\max \sum_{i}\sum_{f \in \mathcal{F}^t_i} \bigg(\mathcal{J}^i(s^i_t,f) +
             \gamma \hat{V}^i(s_t^{i,f}) \bigg) \times a_t^{i,f}$

12. &nbsp;&nbsp;&nbsp;&nbsp;&nbsp;&nbsp;Subject to constraints :


&nbsp;&nbsp;&nbsp;&nbsp;&nbsp;&nbsp;&nbsp;&nbsp;&nbsp;&nbsp;&nbsp;&nbsp;&nbsp;&nbsp;&nbsp;&nbsp;&nbsp;&nbsp;$\sum_{f \in \mathcal{F}_{t}^{i}} a_t^{i,f} = 1 ::: \forall i \in \mathcal{R} $

&nbsp;&nbsp;&nbsp;&nbsp;&nbsp;&nbsp;&nbsp;&nbsp;&nbsp;&nbsp;&nbsp;&nbsp;&nbsp;&nbsp;&nbsp;&nbsp;&nbsp;&nbsp;$\sum_{i \in {\cal R}} \sum_{f \in {\cal F}_t^{i};j \in f} a_t^{i,f} \leq  1 ::: \forall j \in {\cal U} $

&nbsp;&nbsp;&nbsp;&nbsp;&nbsp;&nbsp;&nbsp;&nbsp;&nbsp;&nbsp;&nbsp;&nbsp;&nbsp;&nbsp;&nbsp;&nbsp;&nbsp;&nbsp;$a_t^{i,f} \in \{0,1\} ::: \forall i,f$

13. &nbsp;&nbsp;&nbsp;&nbsp;&nbsp;&nbsp; Assign actions based on the solution of MIP.
14. &nbsp;&nbsp;&nbsp;&nbsp;&nbsp;&nbsp; Simulate agents to pick up requests.

---

> ### Author Response · Authors · 2021-11-18
> **Common Response to all Reviewers (2/2)**
>
> ***Computational Time :***
>
> The recorded wallclock time includes steps 6-13 of Pseudocode Steps 2-4 are one time initialization overheads. Step 5 is a loop over each decision epoch. Steps 6-13 : collect requests, solve the optimization task and assign actions. Step 14 : simulation of the agents, corresponds to the taxis moving physically and picking up or dropping off passengers.
>
> The following points explain the computational complexity of CEVD being at par with NeurADP.
> 1. Since the conditional probability of choosing an action depends only on the distance between action destinations (whose domain is the set of cluster centres), we can pre-compute all possible K $\times$ K values up to a proportionality constant (here K is the number of clusters). Later when computing CEVD values we can fetch these values from the pre-computed matrix. This is a one-time overhead :  Step 3 of the pseudocode.
> 2. To compute CEVD values, apart from computing the individual values, we need a dot product of the conditional probabilites with the individual values of other agents and sum them up for all neighbours in the cluster. ***This is the only minor computational overhead we have over NeurADP at execution time***. Once we have the CEVD values, solving the MIP has the same computational complexity as NeurADP.
>  3. The following table shows the wall-clock time averaged over 60 decision epochs during peak hours. We have the following observations :
>
> &nbsp;&nbsp;&nbsp;&nbsp;&nbsp;&nbsp;A. With the increase in the pickup delay and capacity, the feasible set size increases, thereby increasing the number of actions for each agent and consequently increasing the decision variables per agent over which MIP has to optimize.
>
> &nbsp;&nbsp;&nbsp;&nbsp;&nbsp;&nbsp;B. Compute time increases marginally with number of vehicles ($N$).
>
> &nbsp;&nbsp;&nbsp;&nbsp;&nbsp;&nbsp;C.  As the number of clusters decreases, the number of agents per cluster increases and the overhead talked of in point 2 above increases the time. We note the even with $N=3000$ agents and K=50 clusters the average time per decision epoch is only $3.51$ secs, making our approach deployable in real-time settings. Having K=50 already increases the size of each cluster significantly, thereby requiring to capture dependencies among unrelated agents (i.e., agents which are not competing for the same requests).
>
> | Wallclock time averaging over 60 decision epochs during peak hours (number of requests is highest during peak hours), K denotes the number of clusters. |                 |                |                |                |                |                |
> |--------------|-----------------|----------------|----------------|----------------|----------------|----------------|
> ||Parameters||                                            |Avg. |time |(secs) |
> |Varying| Number of Vehicles       | Pickup Delay       |   Capacity       | K=50 |       K=100  | K=200|
> |Pickup Delay|500 |90 | 4 | 1.86 |1.72 | 1.37|
> ||500| 120 | 4 |1.93 |1.85 |1.53|
> ||500|150|4|2.07|1.99|1.76|
> |Number of Vehicles|500 |90 |4| 1.86 |1.72| 1.37|
> ||750|90|4|2.01|1.81|1.67|
> ||1000|90|4|2.53|2.31|2.03|
> ||2000 |90 |4|2.78 |2.47|2.28|
> ||3000|90|4|3.51|2.89|2.54|
> |Capacity|500 |90 |4| 1.86 |1.72| 1.37|
> ||500|90|5 |1.92|1.77 |1.40|

---

### Decision · Program_Chairs · 2022-01-20

**Decision:**

Reject

**Comment:**

This paper extends a recent approximate dynamic programming method (i.e., DP with neural networks) for a ride sharing problem.
An elegant trick is proposed to obtain a more expressive function approximation without suffering a combinatorial explosion of the action space. While the idea is somewhat ad hoc in its implementation, and limited in novelty w.r.t. the ADP work that the paper builds on, the empirical performance improvement on the ride sharing problem is clear.
Initially, the reviewers also raised several clarity and presentation issues, but the authors did a good job in addressing them in their rebuttal.

The reviewers gave scores of 5,8,5. The main critique is limited novelty.
During the discussion, we focused on the novelty of the approach, whether the ideas can be generalized beyond the very specific ride sharing problem, and whether the work is strong enough if viewed as an application paper.
The conclusion, which my final decision is based on, is that currently, the contribution is very specific to the ride sharing problem, and it is not clear whether this idea can be extended to more general optimization problems. This means that the scope of the algorithmic approach, taken with respect to the ICLR audience, is rather narrow. On the other hand, the current presentation does not meet the bar of a strong application paper, as there is not enough novelty in the problem and data.

My advice to the authors is to broaden their investigation and evaluation. Another option would be to target a venue that is more focused on the ride sharing problem.